# Isolating, identifying and evaluating of oil degradation strains for the air-assisted microbial enhanced oil recovery process

**Mingming Cheng**[1]*, **Long Yu**[2], **Jianbo Gao**[3], **Guanglun Lei**[4], **Zaiwang Zhang**[1]

**1** Institute of Chemical and Safety Engineering, Binzhou University, Binzhou, Shandong, People's Republic of China, **2** Department of Civil & Environmental Engineering, National University of Singapore, Singapore, Singapore, **3** Binzhou Industry and Information Bureau, Binzhou, China, **4** Institute of Petroleum Engineering, China University of Petroleum (East China), Qingdao, Shandong, People's Republic of China

* Rachel19cheng@163.com

**Data Availability Statement:** All relevant data are within the manuscript and its Supporting Information files.

## Abstract

Due to the inefficient reproduction of microorganisms in oxygen-deprived environments of the reservoir, the applications of microbial enhanced oil recovery (MEOR) are restricted. To overcome this problem, a new type of air-assisted MEOR process was investigated. Three compounding oil degradation strains were screened using biochemical experiments. Their performances in bacterial suspensions with different amounts of dissolved oxygen were evaluated. Water flooding, microbial flooding and air-assisted microbial flooding core flow experiments were carried out. Carbon distribution curve of biodegraded oil with different oxygen concentration was determined by chromatographic analysis. The long-chain alkanes are degraded by microorganisms. A simulation model was established to take into account the change in oxygen concentration in the reservoir. The results showed that the optimal dissolved oxygen concentration for microbial growth was 4.5~5.5mg/L. The main oxygen consumption in the reservoir happened in the stationary and declining phases of the microbial growth systems. In order to reduce the oxygen concentration to a safe level, the minimum radius of oxygen consumption was found to be about 145m. These results demonstrate that the air-assisted MEOR process can overcome the shortcomings of traditional microbial flooding techniques. The findings of this study can help for better understanding of microbial enhanced oil recovery and improving the efficiency of microbial oil displacement.

## 1. Introduction

The current rapid pace of economic and social development worldwide necessitates the use of ever-increasing amounts of resources [1]. At the same time. non-renewable resources, such as oil, are at risk of depletion [2], consequently, applying enhanced oil recovery (EOR) techniques in developed fields is crucial to the maximization of scarce oil resources [3]. Microbial enhanced oil recovery (MEOR) is an effective method for improving oil recovery at the late stage of oilfield development [4]. It offers many advantages, such as low cost, powerful adaptability, simple construction, harmless reservoir and no-pollution [5–7]. [8] reported that in

**Funding:** This research was supported by the National Natural Science Foundation of China (Grant No. 51604291), the Natural Science Foundation of Shandong Province (Grant No. ZR2017PEE003), the Natural Science Foundation of Shandong Province (Grant No. 2016ZRB01A57), the Fundamental Research Project of doctor Funded by Binzhou university (Grant 2016Y31), the Seed Fundamental Research Funded by Binzhou university (Grant No. 16CX02010A)

**Competing interests:** The authors have declared that no competing interests exist.

comparison to chemical and physical treatments, microbial treatments may offer a non-hazardous and economically viable strategy for the prevention of paraffin deposition. However, the technique also faces certain challenges, as evidenced by unsuccessful field trials and laboratory experiments [9,10]. The unsatisfactory results were possibly caused by the oxygen-deprived environment inside the reservoir, as well as inappropriate types of bacterial strains [11,12]. Most of the screened strains are aerobic bacteria or facultative anaerobes—that is, they grow fastest under aerobic conditions [13]. Given this, promoting the growth of reservoir microorganisms via air injection is essential for highly profitable long-term performance [14,15]; however, continuous air injection into the reservoir comes with a risk of oxygen breakthrough [16]. In such case, the oxygen concentration may be reduced to a safe level through microorganism growth [17]. There are some shortcomings in the current mathematical models of microbial flooding that microbial growth processes are rarely shown. Most of the classical microbial oil displacement models do not contain microbial growth. The process of model calculation mainly includes the change of metabolite components, such as biosurfactant and biopolymer [18]. Against this backdrop, the present study investigates the potential of the air-assisted MEOR process to overcome some of the shortcomings of traditional MEOR methods, such as a long implementation period. Three oil degradation bacterial strain were isolated from the sludge of the oil producing wells. The strain identification tests revealed that one of the strains was *Pseudomona*, the second strain was *Enterobacter*, the third strain was *Bacillus licheniformis*. This study will help to build a foundation for improving the effects of MEOR. Air-assisted Microbial enhanced oil recovery technology in this study is an economic and effective measure, which is important to the continuous development of constructing well.

The steps of the work were as flows: First, the materials and methods are presented. Then, the strain was identified and the microbial oxygen consumption rate is determined. Then, the influence of dissolved oxygen concentration on the carbon number distribution of biodegraded oil is measured. Then, the physical simulation of air-assisted MEOR in heterogeneous porous media is carried out. Finally, the mathematical model of air-assisted MEOR is established, the air-assisted microbial flooding performance is predicted. The general sketch of this study was shown in Fig 1.

## 2. Materials and methods

### 2.1. Materials

Fermentative enrichment cultures were obtained from the sludge samples near oil wells located in the Triassic sandstones of the Chang-6 reservoir in China. The physical properties of crude oil from this reservoir are shown in Table 1. The cultures were stored at a temperature of 4°C, because the strain is dormant at this temperature [3]. The single colonies with different morphologies were isolated and inoculated on the selected medium [19]. The cultures were grown in a nutrient medium prepared using an artificial brine at typical reservoir temperature (44.4°C) [20,21]. The artificial brine used in the experiments was prepared to match the salinity of the formation water in the Chang-6 reservoir. The total salinity of the artificial brine was 91173 mg/L, with ion composition as shown in Table 2.

### 2.2 Experimental settings

The air-assisted MEOR experiments were conducted using a core flooding apparatus, as illustrated in Fig 2. The apparatus included three-piston intermediate container, pressure sensors, a PL-203 electronic balance, a 101 electric heating incubator, and an air pump, all of which were purchased from Petroleum Research Instrument Co., Ltd., Jiangsu, China. A model porous medium (heterogeneous sand pack) was used in the experiments, the structure of

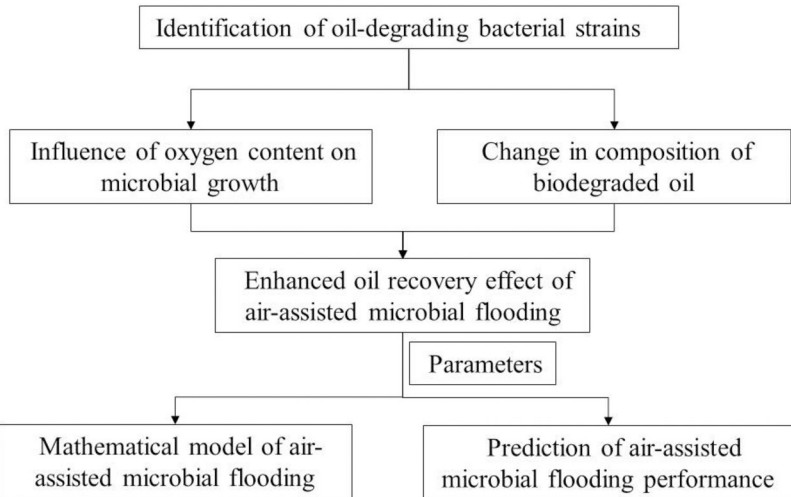

**Fig 1. The general sketch of this study.**

which is shown in Fig 3. The heterogeneous sand models were made by the transparent organic glass. The sand pack tube was composed of a vertical plexiglass tube and a columnar bronze network. The size of vertical plexiglass tube is of diameter 19.5 mm and length 175 mm. The column was packed with different specifications of quartz sand. Higher-permeability sand was packed in the bronze network zone, and lower-permeability sand was packed in the annular space between the bronze network and the pipe inner wall.

## 2.3 Determination of microbial oxygen consumption rate

Most of the production strains that utilizing residual oil as a carbon source are facultative anaerobic bacteria. Dissolved oxygen concentration is a key quality indicator in the microbial growth process [22,23]. Therefore, oxygen consumption rate experiments were conducted to test the effect of air injection on microbial metabolism. The experiment was designed according to the definition of microbial oxygen consumption. Microbial oxygen consumption is the amount of dissolved oxygen in water consumed by microbial reproduction and metabolism. Oxygen consumption rate is the rate at which microorganisms use up oxygen for respiration [23]. The experimental procedure was as follows: 1000 mL of the culture medium was prepared using the artificial brine and injected into three different of 250 mL culture flasks. The dissolved oxygen in the flasks was eliminated by bubbling nitrogen through the solution for 10 minutes. Another flask containing the same volume of the culture media without deoxygenation was used as a control. After sterilization for 30 min in high-pressure steam (0.12MPa), air was introduced into the first three flasks via a gas flow control devices to set the dissolved oxygen concentrations to the desired values of 3.0, 4.5, and 5.5 mg/L. The flasks were then vaccinated with 3 vol% inoculum and incubated at the reservoir temperature, while stirring continuously, for 5 days. At regular intervals of 12 hours, the dissolved oxygen concentration and

**Table 1. Physical property of oil.**

| Viscosity($mPa.S$) | | Freezing point($°C$) |
|---|---|---|
| **Viscosity of reservoir oil** | **Viscosity of gas-free oil** | |
| 1.91 | 14.15 | 20.77 |

**Table 2. Formation water salinity.**

| pH value | Analysis item/$mg\ L^{-1}$ | | | | | | | | | |
|---|---|---|---|---|---|---|---|---|---|---|
| | $CO_3^{2-}$ | $HCO_3^-$ | $Cl^-$ | $SO_4^{2-}$ | $Ba^{2+}$ | $Ca^{2+}$ | $Mg^{2+}$ | $K^++Na^+$ | Salinity | Water type |
| 6.80 | 0 | 80 | 56480 | 0 | 650 | 21000 | 80 | 12980 | 91173 | $CaCL_2$ |

the colony density in the solution were monitored. The oxygen consumption rate was calculated by combining the dissolved oxygen concentration and the colony density. The experimental setup is illustrated in Fig 4.

## 2.4 Influence of dissolved oxygen concentration on the carbon number distribution

In MEOR, the microorganisms introduced into the reservoir utilize the long chain hydrocarbons of residual oil as a carbon source. In the present study, gas chromatography was used to measure the influence of the dissolved oxygen concentration on the carbon number distribution of biodegraded oil. The experimental procedure was as follows: the culture media

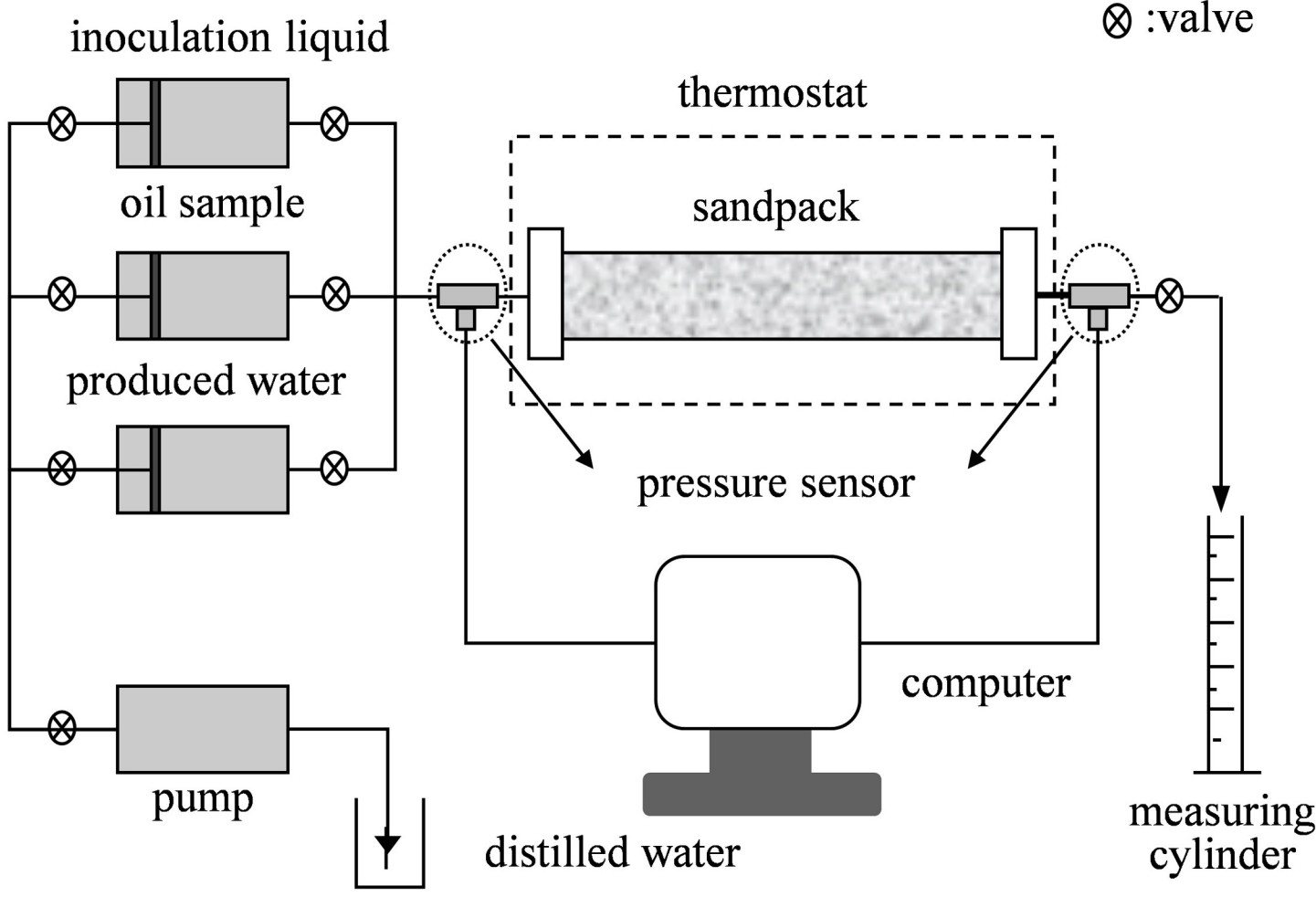

**Fig 2. Core flooding experiment model.**

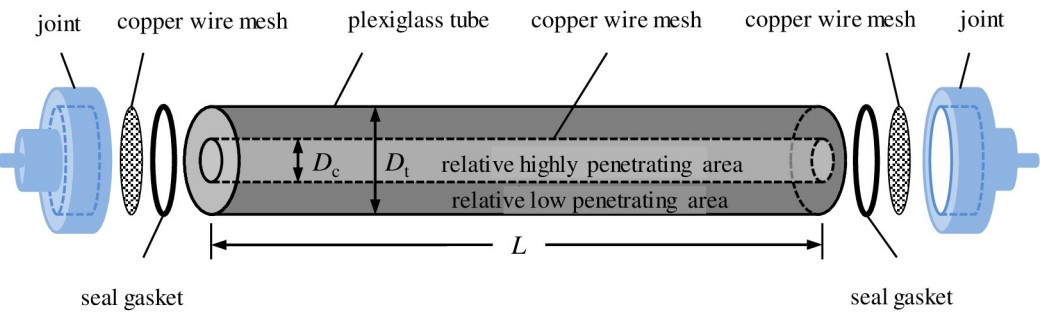

**Fig 3. Heterogeneous model of the single pipe.**

prepared using the artificial brine were injected into two different culture flasks, in which the dissolved oxygen concentrations were set as 0.5 mg/L and 5.5 mg/L, respectively. After sterilization in high-pressure steam for 30 min, the flasks were vaccinated with 3 vol% inoculum, following by injection with crude oil and incubation at reservoir temperature, while stirring continuously, for 5 days. A third flask, containing the same amount of brine medium but not the experimental strain, was used as a control. A combination of gas chromatography method and quantitative analysis was used to determine the change in composition of the oil.

## 2.5 Physical simulation of air- assisted MEOR in heterogeneous porous media

Core flood experiments were conducted to test the oil recovery effect of air-assisted microbial flooding in heterogeneous porous media. Prior to the experiments, the sand pack columns were put under vacuum pressure and saturated with artificial brine to measure their permeability and porosity. A model oil (mixture of crude oil and kerosene) was viscosity-matched with the reservoir oil (the viscosity at reservoir temperature was 1.91 MPa·s) and was used for

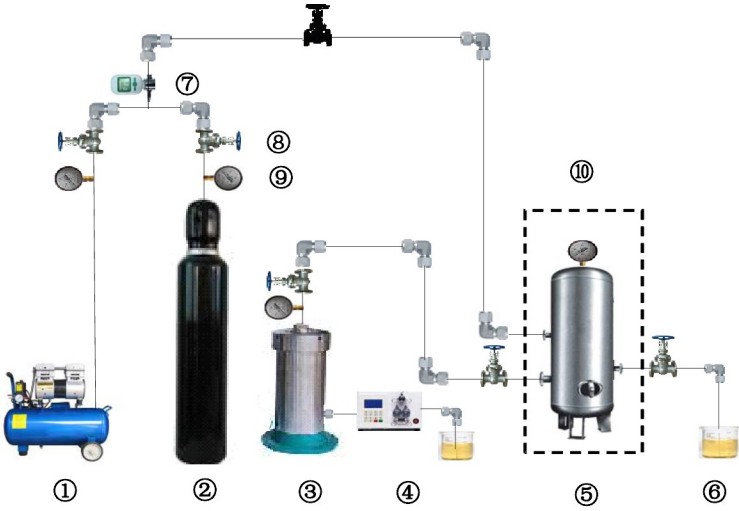

tips: ①air pump; ②nitrogen cylinder; ③bacteria solution; ④constant-flux pump; ⑤incubator; ⑥Liquid sample; ⑦gas flowmeter; ⑧ control valves; ⑨pressure meter; ⑩constant temperature oven

**Fig 4. The experimental facilities of microbial oxygen consumption rate determination.**

the flooding experiments. Prior to use, the model oil mixture was filter-sterilized using a 0.22 μm filter. To create the initial core conditions for the flooding experiments, the sand pack columns were saturated with model oil, then flooded with brine until the water cut at the outlet reached 60–75%. The columns were alternately inoculated with bacteria suspension and air (except for one column, which was used a control). The columns were then shut in and incubated at the reservoir temperature for 3 days, before being flooded with brine until no further oil was discharged at the outlet.

## 3. Mathematical model of air-assisted MEOR

Under the precondition of incompressible flow, the capillary pressure and gravity are not considered, the mathematical model used in this study is based on the Buckley-Leverett theory and Darcy's law. The calculation steps are shown in Fig 5.

### 3.1 The oxygen-consumption model

As the oxygen consumption during the low-temperature oxidation of oil in a high water-cut and low-temperature reservoir is much smaller compared to the microbial consumption, it was assumed for the purpose of this study that the relationship between microbial growth and oxygen consumption velocity in the laboratory experiments was representative of the reservoir. Consequently, only the microbial oxygen consumption is considered here. On account of the high porosity in the model porous medium, the strata showed a large specific surface characteristic. Therefore, the injected inoculum and air were considered to be fully dispersed in the pores. And as such, the high reservoir pressure would impel the injected air to disperse even better in the bacteria solution. Concrete method for oxygen-consumption model was shown in Fig 5 [1,10].

For simplicity, we assume that the viscosity of crude oil is $\mu_o$, the viscosity of aqueous phase is $\mu_w$, oil relative permeability is $K_{ro}$, water relative permeability is $K_{rw}$. After the microbial activity and air injection, oil viscosity turns into $\mu_{om}$, and oil relative permeability turns into $K_{rom}$.

Assuming the viscosity reducing rate of reservoir oil was the same with the stock tank oil, an estimation of the impact of air-assisted microbial activity on reservoir oil viscosity was

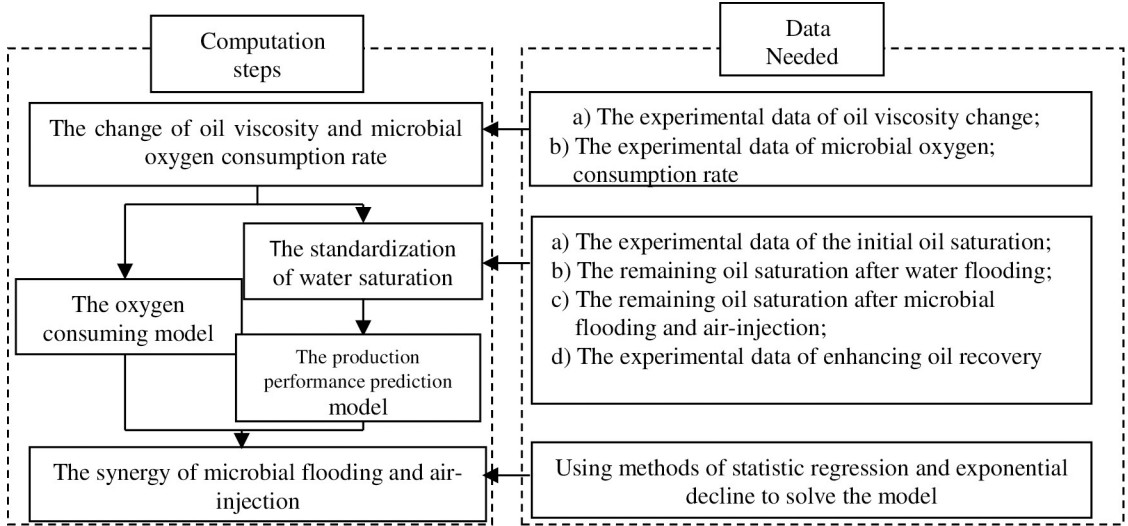

**Fig 5. Concrete method for oxygen-consumption model.**

proposed. The viscosity of microbial degraded oil, $u_{om}$, may calculated by Eq (1) [10]:

$$\mu_{om} = \mu_o(1 - n) \tag{1}$$

where n is the viscosity reduction rate of degassed oil.

Based on the experimental statistical regression, it is believed that the oil and water relative permeability in the water-wet reservoir, $k_{ro}$, $k_{rw}$, and the standardization of water saturation, $S_{wD}$, may be calculated by the relevant empirical Eqs (2), (3) and (4) [10]:

$$k_{ro} = \lambda(1 - S_{wD})^{\varepsilon} \tag{2}$$

$$k_{rw} = \omega S_{wD}{}^{\delta} \tag{3}$$

$$S_{wD} = \frac{S_w - S_{wi}}{1 - S_{wi} - S_{or}} \tag{4}$$

where $S_w$ is the water saturation, $S_{or}$ is the residual oil saturation, $S_{wi}$ is the irreducible water saturation, $\lambda$ is the oil phase relative permeability when $S_w = S_{wi}$, $S_{wD} = 0$; $\omega$ is the water phase relative permeability when $S_w = 1-S_{or}$, $S_{wD} = 1$; $\varepsilon$ and $\delta$ are constants depend on rock wettability and characteristics of pore structure [10].

The residual oil saturation after air-assisted microbial flooding $S_{or}$ may be calculated by Eq (5):

$$S_{or} = S'_{or} - S_{oi}(\Delta E_R) \tag{5}$$

where, $\Delta E_R$ is the enhancement of oil recovery, $S_{oi}$ is the initial oil saturation, $S'_{or}$ is the remaining oil saturation after water flooding.

The normalized water saturation $S_{wD}$ may be calculated the data of using the residual oil saturation data of the microbial flooding process, and the correlation coefficients $\lambda$, $\omega$,$\varepsilon$ and $\delta$ may be calculated by fitting a curve using Eqs (3) and (4). The oil-water relative permeability equation for a water-wet reservoir can be derived under the assumption of constant water viscosity.

The injected microorganisms, the breeding cycle of which was up to 20–30 mins, multiplied rapidly under the presence of ample nutrition and air in the reservoir. It was assumed that the microorganisms at the bottom of the well were in a stable period of the growth cycle. The cells concentration of stationary phase in the region reaches a maximum. The maximum oxygen concentration, $c_{max}$, may be calculated using Eq (6):

$$c_{max} = \frac{Q_g T_g \rho_g \sigma_g}{Q_w T_w} \tag{6}$$

where $Q_g$ is the intensity of injectied air,(Sm$^3$/d); $T_g$ is the air injection time over a test cycle, (d); $\rho_g$ is the air density, (g/Sm$^3$); $\sigma_g$ is the mass ratio of oxygen in the air; $Q_w$ is the water injection strength, (m$^3$/d); and $T_w$ is inoculum injection over a test cycle,(d);

The primary oxygen concentration in the microbial stationary phase, $c_0$, may be calculated using Eq (7):

$$c_0 = c_{max} - c_s \tag{7}$$

where $c_s$ is the oxygen consumption in the stationary phase, mg/L;

It is assumed that sufficient nutrition was injected during microbial flooding to ensure that the number of cells consumed in the stationary phase were replenished by the growth of new microorganisms. On account of the pore nearby the injection well was filled with

water during the high water-cut period, the change in oxygen concentration, $c$, may be calculated using Eq (8):

$$\mathrm{d}c = \frac{2\pi\gamma h\phi r}{Q_w}\,\mathrm{d}r \qquad (8)$$

where $\gamma$ is the total oxygen consumption rate in the stationary phase (g/(m$^3$·d)), $h$ is the reservoir thickness (m), $\phi$ is the porosity of the oil layer, and $r$ is the radius of the oil layer (m).

The oxygen concentration as a function of the radial coordinate $r$, $c(r)$, may be calculated using Eq (9).

$$c(r) = c_0 - \frac{\pi\gamma h\phi r^2}{Q_w} \qquad (9)$$

In an actual reservoir, most of the microbial oxygen consumption occurs near the wellbore [24,25]. The reduced microbial number is replenished through the continuous addition of sufficient nutrients in the injection water during microbial flooding near the wellbore.

In an actual reservoir, most of the microbial oxygen consumption occurs near the wellbore [24,25]. In such case, the reduced microbial number is replenished through the continuous addition of sufficient nutrients in the injection water during microbial flooding.

The oxygen consumption in the logarithmic phase, $C_m$, was calculated using Eq (10):

$$C_m = C_s - C_o \qquad (10)$$

where $C_s$ is the initial dissolved oxygen concentration, (mg/L): $C_0$ is the dissolved oxygen concentration corresponding to the maximum cells concentration, (mg/L).

The metabolic oxygen consumption rate in the stationary and declining phases, $\gamma_2$, was calculated by fitting an oxygen consumption curve, according to Eq (11):

$$\gamma_2 = \frac{C_m}{C_{m\,max}} \times \frac{C_{m\,max} - C_{m\,e}}{t_2 - t_1} \qquad (11)$$

where $C_{me}$ is the microbial concentration at the end of the experiment ($10^8$ cells/mL), $t_1$ is the time at which the maximum microbial concentration was attained (d), and $t_2$ is the end time of the experiment (d).

## 3.2 Model for predicting pilot site performance

The predictive model for oil recovery was built according to the well location and permeability of the pilot site (Fig 6). The mesh dimensions were $110 \times 105 \times 14$, and the total grid number was 161700, which included 60662 effective grids.

According to the well pattern design of the pilot site, there are 22 wells on the block. The wells area began production in 1990. In the beginning, all 22 were production wells, with each individual well having a liquid production capacity of 2.73 m$^3$/d. After 6 months following the start of production, the Wang 18–2 and Wang 19–2 wells were changed from production wells into injection wells, with an injected water volume of 10 m$^3$/d. After a further 6 months, the Wang 20–1 and Wang 21–3 wells were also changed from production well into injection well, with an injected water volume of 20 m$^3$/d. The other wells kept producing at a liquid production capacity of 3.33m$^3$/d. The main simulation parameters are shown in Table 3.

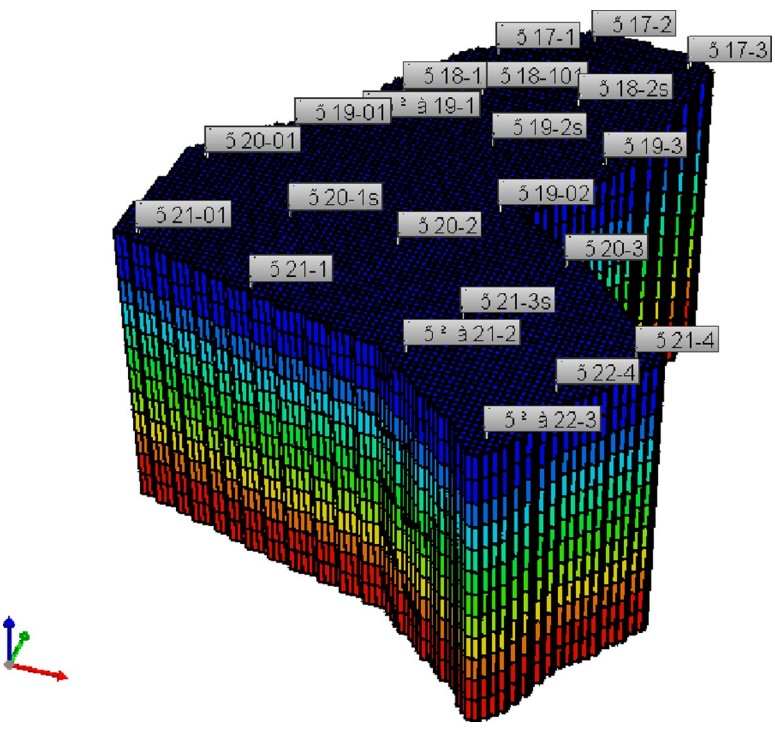

**Fig 6. Conceptual geological model.**

## 4. Results and discussion

### 4.1 Identification of oil-degrading bacterial strains

Three oil degradation bacterial strains, Cq3-1, Cq3-2, and Cq3-3 were isolated from the sludge of the oil producing wells. The strain characteristics were identified, and 16S rDNA sequence analysis was performed using primers containing multiple restriction enzyme sites to amplify the strain DNA. The clear and bright stripes of amplification products were chosen and sequenced, the sequencing results were searched similarity in the database, the last result of the tested bacteria strains' 16S rDNA was shown in Table 4. The higher homologous sequences were used to construct a phylogeny tree (Fig 7). The digits represent the similarity of the strains to the database cultures.

The strain identification tests revealed that one of the strains was *Pseudomonas*, which is a type of aerobic bacteria. This strain has a strong ability to break down organics into molecules such as glycolipids and lipopeptides [10,25]. The second strain was *Enterobacter*, which is a typical anaerobic bacterium and reproduces rapidly utilizing simple organic

**Table 3. Main parameters of simulation.**

| name of parameter | value | name of parameter | value |
|---|---|---|---|
| depth of oil layer /m | 1200 | average permeability /($10^{-3}\mu m^2$) | 3.3 |
| effective porosity /% | 13.25 | initial formation pressure /MPa | 9.13 |
| initial oil saturation/% | 56 | formation temperature /°C | 48.5 |
| underground crude oil density /(g·cm$^{-3}$) | 0.754 | underground crude oil viscosity /(mPa·s) | 1.96 |
| bulk coefficient | 1.21 | coefficient of compressibility /($10^{-4}$·MPa$^{-1}$) | 10.8 |
| produced crude oil viscosity /(mPa·s) | 4.85 | produced crude oil density /(g·cm$^{-3}$) | 0.8403 |

**Table 4. Blast result of the tested bacteria strains′16S rDNA.**

| Tested bacteria Strain | | Reference strain | | Homology/% | Identification result |
|---|---|---|---|---|---|
| Strain name | Accession No. | Genus species | Accession No. | | |
| Cq 3–1 | KJ782614 | *Pseudomonas veronii* CIP 104663 | CIP 104663 | 99.93 | *Pseudomonas* |
| Cq 3–2 | KJ782615 | *Enterobacter xiangfangenis* 10–17 | 10–17 | 99.78 | *Enterobacter xiangfangenis* |
| Cq 3–3 | KJ782616 | *Bacillus licheniformis* ATCC 14580 | ATCC 14580 | 98.86 | *Bacillus licheniformis* |

compounds as a carbon source. It uses two main methods for microbial sugar fermentation, one of which leads to mixed organic acids such as formic acid, acetic acid, and succinic acid as its final products. The third strain was *Bacillus licheniformis*, which is a typical facultative anaerobe. This strain tends to be strongly temperature resistant as well as alkali and acid proof. Of the three bacterial strains, *Enterobacter* multiplies rapidly under hypoxic conditions, whereas *Pseudomonas* and *Bacillus licheniformis* multiply rapidly under an oxygen-rich atmosphere.

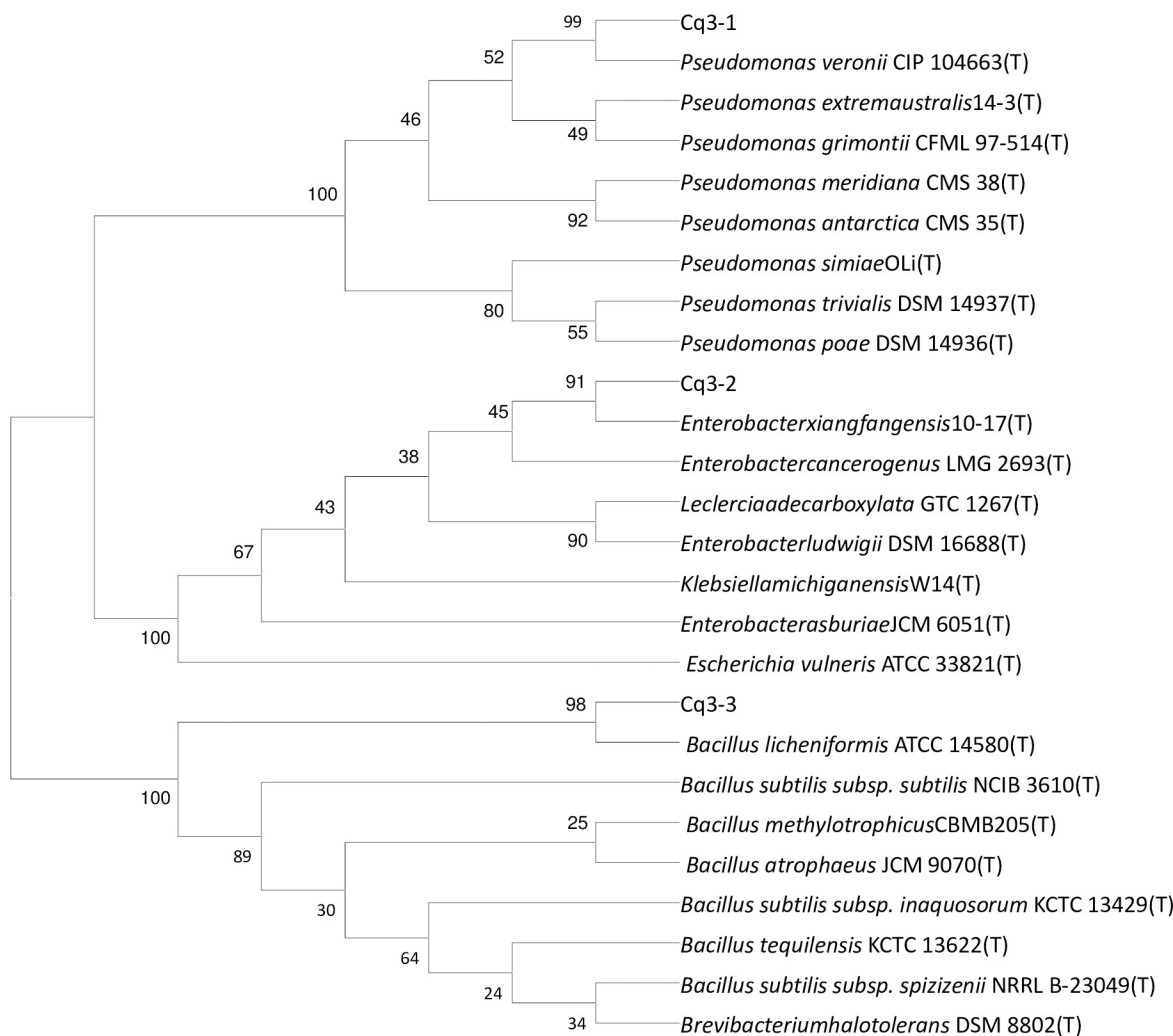

**Fig 7. Evolutionary tree of the relationships among reference strains and tested strains based on 16S rDNA.**

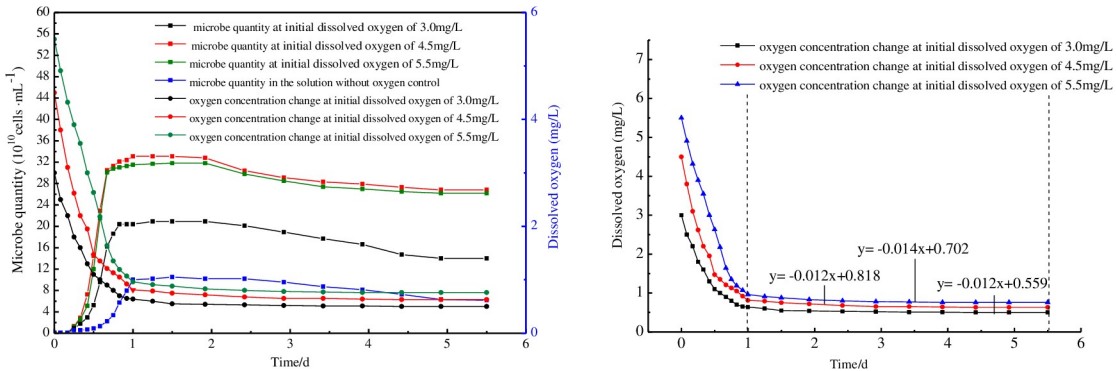

**Fig 8. Strains growth and oxygen concentration change under different initial dissolved oxygen condition.**

## 4.2 Influence of oxygen on microbial growth and reproduction

The cells numbers of the mixed strains in different oxygen concentrations were determined via cell counting analysis, the results of which are shown in Fig 8.

Fig 8A shows that the experimental strains were able to grow in the artificial brine that was used as a control, reaching a maximum cell concentration of $1 \times 10^9$ cells/mL. In comparison, the maximum cell concentrations in the solutions with 4.5 and 5.5 mg/L of dissolved oxygen were at $3.40 \times 10^9$ cells/mL and $3.35 \times 10^9$ cells/mL respectively, which were three times higher than that of the control. The cell growth curve of artificial brine showed an obvious lag phase. These data support the hypothesis that there is an optimum concentration of dissolved oxygen which is most conducive to microbial growth and reproduction. The microbial oxygen consumption curves followed the same trend for all the solutions, namely that the dissolved oxygen concentration decreased logarithmically in the logarithmic growth phase and linearly in the stationary and declining phases [26].

The parameters for the oxygen concentration change equation can be obtained by fitting to the test data (Fig 7B). The metabolic oxygen consumption rates corresponding to 3.0, 4.5, and 5.5 mg/L of oxygen concentration were 0.012, 0.014, and 0.012 mg/(L·d), respectively. A combined least squares curve-fitting and difference calculation method was used to analyze the oxygen consumption rate, with the results shown in Table 5. Table 5 shows that the oxygen consumption rate of microorganism is different with different oxygen content. This is because the metabolic rate of microorganisms was different under different oxygen content. The oxygen concentration distribution was calculated using the data of microbial metabolic rate.

## 4.3 Change in composition of biodegraded oil at different dissolved oxygen concentrations

The changes in composition of the biodegraded oil at different dissolved oxygen concentrations are shown in Fig 9.

**Table 5. The microbial experimental data processing and results.**

| $C_s$/ (mg·L$^{-1}$) | $C_{mmax}$/(10$^8$cells·mL$^{-1}$) | $t_1$/d | $C_o$/ (mg·L$^{-1}$) | $C_{me}$/ (10$^8$cells·mL$^{-1}$) | $t_2$/d | $C_m$/ (mg·L$^{-1}$) | $\gamma_1$/ (mg·L$^{-1}$·d$^{-1}$) | $\gamma_2$/ (mg·L$^{-1}$·d$^{-1}$) | $\gamma$/ (mg·L$^{-1}$·d$^{-1}$) |
|---|---|---|---|---|---|---|---|---|---|
| 3.0 | 20.9 | 1.5 | 0.55 | 14 | 5.5 | 2.45 | 0.012 | 0.2022 | 0.2142 |
| 4.5 | 34.0 | 1.5 | 0.68 | 26.8 | 5.5 | 3.82 | 0.014 | 0.2022 | 0.2164 |
| 5.5 | 33.5 | 1.5 | 0.81 | 26.2 | 5.5 | 4.69 | 0.012 | 0.2555 | 0.2675 |

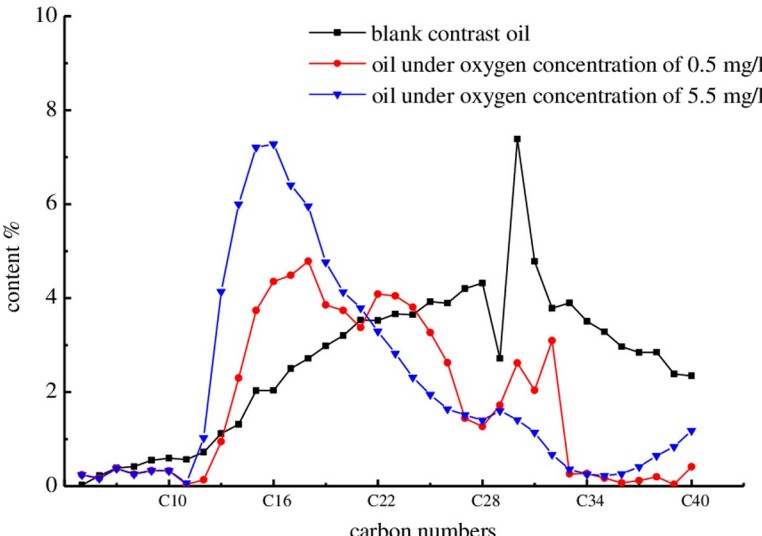

**Fig 9. Carbon distribution curve of biodegraded oil with different oxygen concentration.**

As Fig 9 shows, the efficiency of microbial oil degradation is positively correlated with the dissolved oxygen concentration. The C15 fraction of the oil that was incubated with 5.5 mg/L of dissolved oxygen increased by about 5.18%, while its C30 fraction decreased by about 5.98%. By contrast, the corresponding numbers for the oil incubated with 0.5 mg/L of dissolved oxygen were 1.71% and 4.77%, respectively. A likely explanation for this is that the aerobic and facultative bacteria utilize residual oil as a carbon source in an oxygenated environment. Oxygen atoms are introduced in the long-chain alkanes by alkane oxygenases through terminal and penultimate oxidation. The long-chain alkanes are then oxidized into the corresponding alcohols, aldehydes, and fatty acids. This process is very conducive to lowering the oil viscosity.

## 4.4 Enhanced oil recovery effect of air-assisted microbial flooding

Core flooding experiments were required to investigate the complementary effects of microbial flooding and air injection, with the results shown in Table 6.

Core flooding experiments were conducted using artificial cores, thus the parameters of artificial cores could not be exactly consistent. The permeability, porosity and pressure of artificial cores were as close as possible. As Table 6 shows, the use of diverse flooding methods made a significant difference to the enhanced oil recovery effects. A likely explanation for this

**Table 6. Main results of microbial oil displacement experiment.**

| No | average permeability /(10$^{-3}$μm$^2$) | porosity | oil saturation/ % | microbial injection quantity | Injection pattern | injective time | final pressure /MPa | pressure drops /% | recovery before injection % | final recovery / % | Improvement of recovery /% |
|----|------|------|-------|------|------|------|------|------|------|------|------|
| 1 | 180 | 0.42 | 73.85 | 0 | water flooding | / | 47 | | 0 | 23.37 | / |
| 2 | 200 | 0.39 | 69.51 | 0.3PV | Two slug of microbial flooding | $f_w$ = 79% | 25 | 46.81 | 18.14 | 31.68 | 8.31 |
| 3 | 220 | 0.38 | 67.16 | 0.3PV | air-microorganism-air- microorganism | $f_w$ = 80% | 20 | 57.45 | 19.91 | 36.64 | 13.27 |
| 4 | 190 | 0.40 | 78.30 | 0.3PV | microorganism-air-microorganism-air | $f_w$ = 82% | 15 | 68.09 | 21.12 | 40.45 | 17.08 |

is that the fingering along the high permeability layer of water flooding formed a dominant flow channel, which reduced the sweep efficiency. The oil recovery of air-assisted microbial flooding was greatly enhanced relative to water flooding and microbial flooding. For example, the enhancements in oil recovery from microbial flooding, air-microorganism-air-microorganism flooding, and microorganism-air-microorganism-air flooding were 8.31%, 13.27%, and 17.08%, respectively. The most likely reason for this is that the injected microorganisms were able to effectively prevent air channeling from occurring along the cracks of the high permeability layer. The relationships between the injection pore volume and the oil recovery, water cut, and injection pressure are shown in Fig 10.

During water flooding, the oil was mainly displaced before water breakthrough occurred, leading to a rapid increase in water cut following breakthrough. The injection pressure gradually increased and reached a maximum value, at which point the injected water reached a certain depth in the core. Following breakthrough, however, the injection pressure decreased rapidly. These data support the hypothesis that the dominant flow channels formed by water flooding give rise to waste displacing fluid. This outcome required more efficient oil displacement technology (Fig 10A). The water cut dropped rapidly following microbial flooding, indicating that the microbial action had an effect on starting residual oil recovery. The injection pressures of the microbial flooding and the subsequent water flooding were smaller than those of the pre-shut-in period. Hence, the results show that the injection capacity was enhanced by microbial action. Taking this one step further, the oil droplets in the pore spaces migrated and formed a strap under the effect

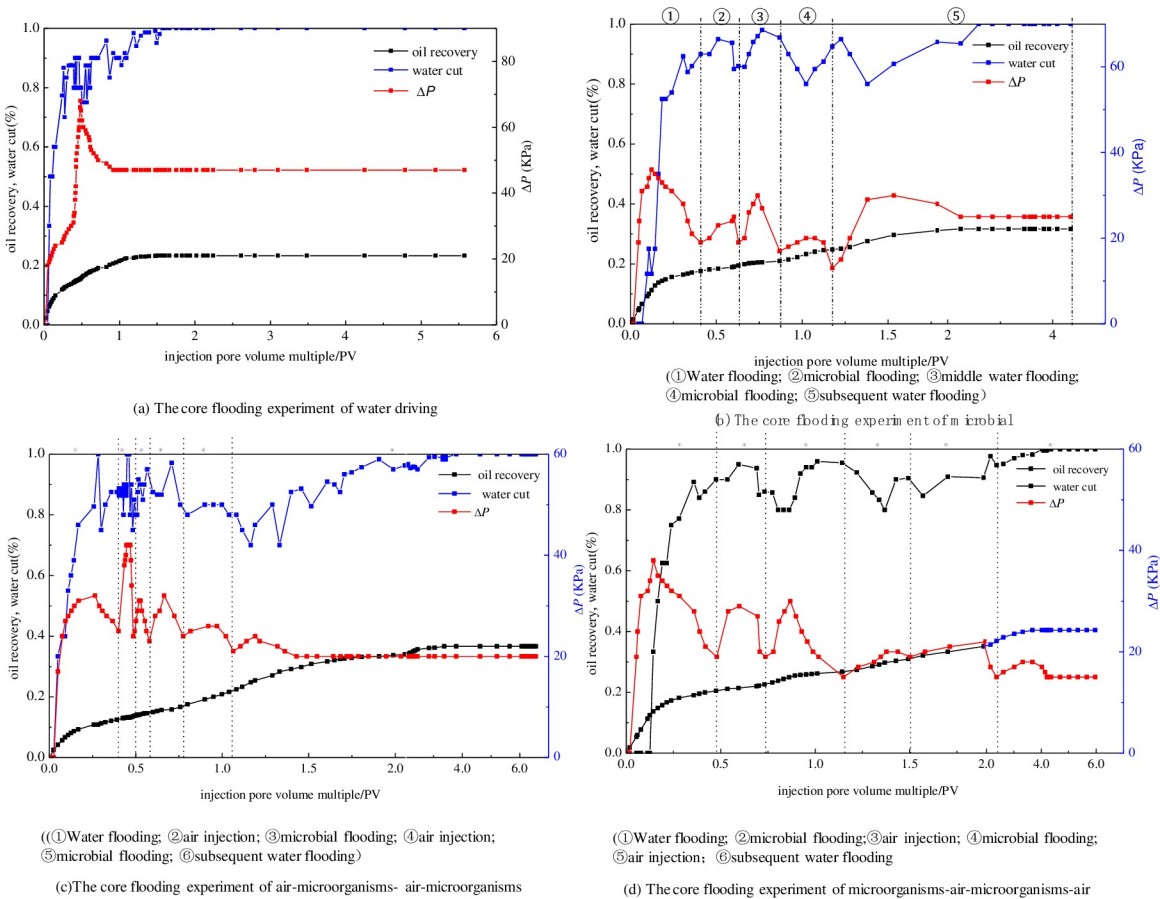

(a) The core flooding experiment of water driving

(①Water flooding; ②microbial flooding; ③middle water flooding; ④microbial flooding; ⑤subsequent water flooding)

(b) The core flooding experiment of microbial

((①Water flooding; ②air injection; ③microbial flooding; ④air injection; ⑤microbial flooding; ⑥subsequent water flooding)

(c) The core flooding experiment of air-microorganisms- air-microorganisms

(①Water flooding; ②microbial flooding; ③air injection; ④microbial flooding; ⑤air injection; ⑥subsequent water flooding

(d) The core flooding experiment of microorganisms-air-microorganisms-air

**Fig 10. The core flooding experiment of water driving under the condition of heterogeneity.**

of various MEOR mechanisms, such as oil-water interfacial tension reduction via biosurfactant adsorption at the interface and oil viscosity reduction via the dissolution of biogas (Fig 10B).

As illustrated in Fig 10C, the water cut fluctuated greatly after microbial injection, reflecting the effect of air injection on cell propagation. Following the injection of microorganisms and air, the swept area extended gradually to each side of the original dominant flow channel. The oil recovery in the air-microorganisms-air-microorganisms injection mode increased significantly at the later stages relative to microbial flooding. The injection pressure sharply increased, to an even higher value than the pressure before water breakthrough, at the beginning of the air injection process. It is conceivable that the injection air was dispersed throughout the core in the form of bubbles, some of which may have obstructed pore throats, producing an air-lock which slowed down the movement of microbes and caused the aforementioned pressure rise. Fig 10D shows that in the microorganisms-air-microorganisms-air injection mode, the pressure was at its maximum value before water breakthrough. While the pressure did fluctuate following microbial injection, its magnitude was far less than that of injecting air followed by microorganisms. This implies that the microbial flooding had an anti-gas channeling effect on air injection. The higher recovery efficiency of the microorganisms-air-microorganisms-air injection mode reflects the synergistic effect of air profile control and high-efficiency microbial flooding. The injection of the microorganisms-air system causes the biopolymer to block the dominant flow channels and adjust the seepage flow profile. The bio-surfactant displaces the columnar and membranous residual oil via emulsification and wettability reversal. Although the water cut fluctuates in some moment, it shows a larger upward trend. The water cut curves of air-assisted microbial flooding have obvious fall during secondary water drive stage. It shows that the air-assisted microbial flooding is effective in water cut control during enhanced oil recovery process. A similar oil recovery properties have been reported for several species of Bacilli bacteria used in core flood experiments [27].

## 4.5 Microbial oxygen consumption and relative permeability results

The coefficients $\lambda$, $\omega$, $\varepsilon$, $\delta$, and $r$ were calculated via Eqs (3) and (4) using the relative permeability data from the field trials. The mathematical models of air-assisted microbial flooding were obtained using a semi-theoretical semi-empirical method [10]. The fitted core was 17.5 cm long with a cross-sectional diameter of around 1.95 cm. The other fitting parameters were set according to the experimental results. The results for water cut, oil recovery, and relative permeability are shown in Fig 11.

The calculation parameters were as follows: reservoir thickness of 13.3 m, porosity of 0.137, alternating injection cycle of 6 months (of which the air injection time was 1 month), water injection time of 5 months, and injection intensity of 50 $m^3$/d. The oxygen consumption rates in the logarithmic, stationary and decline phases were taken as the average values from the experiments. As illustrated in Fig 11, the predictive model showed very good agreement with the measured values. The fitted relative permeability values were consistent with those calculated using the experimental data.

The safe oxygen concentration of air is that for which no explosion occurs when it is mixed with an inflammable gas such as methane [25]. It is known from the literature search that the safe oxygen percentage of air 8% [28], there would be an oxygen concentration corresponding to an air injection intensity. The safe oxygen concentration distribution curve and the oxygen concentration change curve intersect at a point.

As illustrated in Fig 11, the computation showed that the results of this prediction model fit the measured values very well. The fitted relative permeability was consistent with the relative permeability calculated by experimental data.

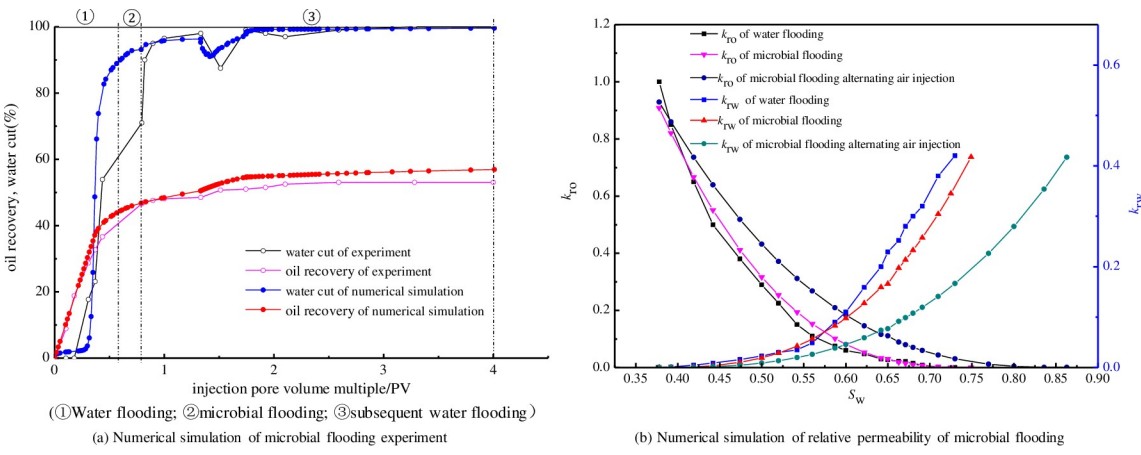

(a) Numerical simulation of microbial flooding experiment

(b) Numerical simulation of relative permeability of microbial flooding

**Fig 11. Numerical simulation of oil recovery and relative permeability of microbial flooding.**

Fig 12A shows the initial oxygen concentration increased with increasing air injection intensity. For air injection intensities of 3000, 4000, and 5000 $m^3$/d, the corresponding oxygen concentrations through microbial oxygen consumption in a radius of 165 m were 64.7, 1720.7, and 2476.7 g/$m^3$, respectively. This condition may still satisfy the requirements of microbial growth and metabolism, thus forming a favorable environment for bacteria incubation. In the meantime, microbial growth and metabolism continuously reduced the oxygen concentration, further improving the safety level of the displaced air. For air injection intensities of 3000, 4000, and 5000 $m^3$/d, the safe oxygen concentrations through microbial oxygen consumption were 1400, 1950, and 2400 g/$m^3$, respectively, with corresponding minimum safe radii for microbial oxygen consumption of 125, 149, and 160 m, respectively. When reservoir radius is greater than the minimum safe radius, the oxygen concentration is below the safe limit. However, taking into account the oxygen consumption under low temperatures would further decrease the minimum safe radius. When the air injection intensity was maintained at 4000 $m^3$/d, increasing the water injection volume reduces the oxygen concentration. The minimum safe radii corresponding to water injection rates of 30, 40, and 50 $m^3$/d were 121, 152, and 188 m, respectively (Fig 12B).

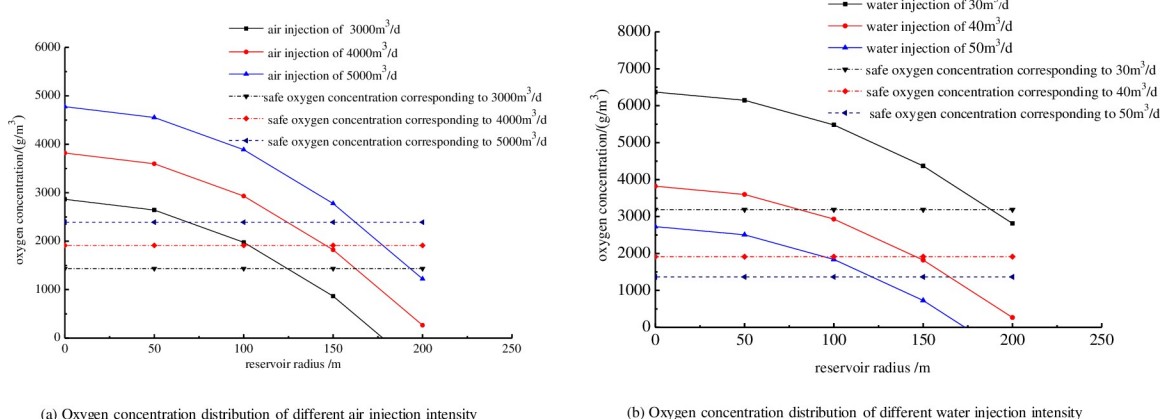

(a) Oxygen concentration distribution of different air injection intensity

(b) Oxygen concentration distribution of different water injection intensity

**Fig 12. Oxygen concentration distribution of different injection intensity.**

### 4.6 Prediction of air-assisted microbial flooding performance

The production performance was simulated, a better developing solution was proposed that the microbial slug at 0.075PV, microorganisms and air being injected a mouth respectively at each cycle (2 months). The microorganisms and air were injected alternately into the four injection wells described in section 3.2. The simulation results showed that this would result in a water cut decrease of 7.26%, an oil production increase of about $4.51 \times 10^4 \, m^3$, and a recovery enhancement of about 7.72% after air-assisted microbial flooding in the test site. The predicted water cut and oil recovery are shown in Fig 13. As the figure shows, the oil recovery of recommendations was apparently higher than the original water flooding. The most probable reason for this is that improving the anoxic formation environment by air injection increases the microbial growth and reproduction efficiency.

The water cut of air-assisted microbial flooding presented a cone of depression, afterward, showed a gradual increase relative to the original water flooding, which indicates synergy between the air injection and microbial flooding.

## 5. Summary and conclusions

A mixture of aerobic (*Pseudomonas*), facultative (*Enterobacter xiangfangenis*) and anaerobic (*Bacillus licheniformis*) bacteria were used in the experiments conducted in this study. The performance of the oil degradation strains for the air-assisted Microbial Enhanced Oil Recovery Process was evaluated, as summarized below.

1. Changes in oxygen concentration showed a significant impact on the microbial growth rate and the maximum microbial proliferation. The maximum cell concentration of the solution incubated with 5.5 mg/L of dissolved oxygen was 2–3 times higher than that of the solution without injected oxygen.

2. The measured microbial oxygen consumption rate was 0.2142–0.2675 mg/(L·d). In addition, most of the microbial oxygen consumption in the reservoir occurred during the stationary and declining phases due to the combined effects of metabolic and growth oxygen consumption.

3. The oil recovery varied with the injection type, with water flooding showing the lowest enhancement performance, microbial flooding being in the middle, and microbe-alternating-air flooding being the best-performing. In the microorganisms-air injection, the air

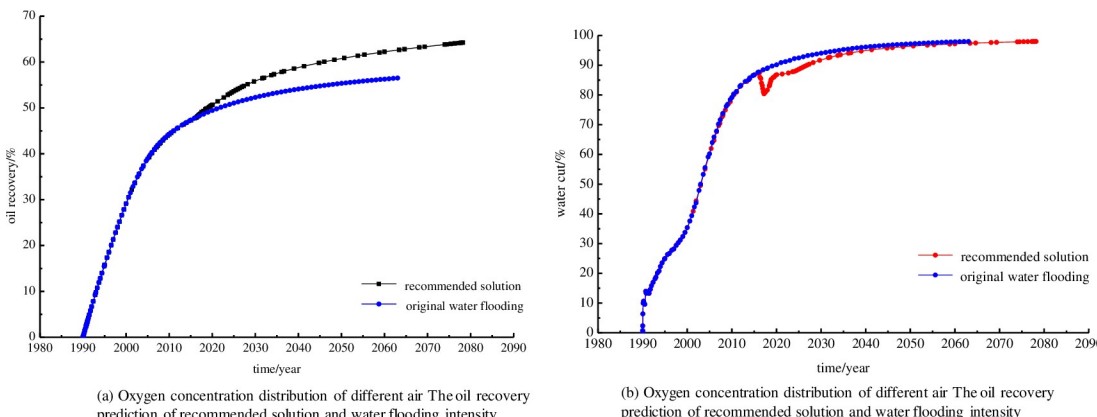

(a) Oxygen concentration distribution of different air The oil recovery prediction of recommended solution and water flooding intensity

(b) Oxygen concentration distribution of different air The oil recovery prediction of recommended solution and water flooding intensity

**Fig 13. The oil recovery and water cut prediction of recommended solution and water flooding.**

bubbles increased the resistance of the pore paths via the Jamin effect, while the oil displacement efficiency was significantly enhanced through the dual mechanisms of profile control and oil displacement.

4. The relative permeability curves for air-assisted microbial flooding were fitted using a mathematical model, the results of which showed very good agreement with experiments. The model results also showed that the oil recovery from microbe-alternating-air flooding was higher than with just water flooding. The most probable reason for this is that the air injection improved the anoxic formation environment, giving rise to improvements in the microbial growth and reproduction efficiency.

5. The microbial oxygen consumption in the microorganisms-air system also reduced the oxygen concentration in the reservoir. When the air injection intensities were 3000, 4000, and 5000 $m^3$/d, the oxygen concentrations in the radius of 165 m through microbial oxygen consumption were 64.7, 1720.7, and 2476.7 $g/m^3$, respectively. The minimum safe radii corresponding to 30, 40, and 50 $m^3$/d of water injection were 121, 152, and 188 m, respectively. The oxygen concentration in these radii can still meet the demands of microbial growth while maintaining the safety of the air injection displacement process.

## Supporting information

**S1 Data.**
(DOCX)

## Author Contributions

**Data curation:** Mingming Cheng, Long Yu.

**Investigation:** Mingming Cheng, Zaiwang Zhang.

**Methodology:** Mingming Cheng.

**Supervision:** Guanglun Lei.

**Writing – original draft:** Mingming Cheng.

**Writing – review & editing:** Jianbo Gao.

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
