## [Decision Letter · Decision Letter 0]

30 Jun 2020

PONE-D-20-04914

Isolating, Identifying, Separating and Evaluating Oil Degradation Strains for the Air-assisted Microbial Enhanced Oil Recovery Process

PLOS ONE

Dear Dr. Cheng,

Thank you for submitting your manuscript to PLOS ONE. After careful consideration, we feel that it has merit but does not fully meet PLOS ONE’s publication criteria as it currently stands. Therefore, we invite you to submit a revised version of the manuscript that addresses the points raised during the review process.

We look forward to receiving your revised manuscript.

Kind regards,

Omeid Rahmani, Ph.D

Academic Editor

PLOS ONE

Additional Editor Comments:

I note that Reviewer 1 has requested that you cite several papers as part of your revisions. There is no requirement from the journal that you cite all of these specific papers, so please feel free to choose only those that are genuinely necessary to provide appropriate context to your study. Also, you don't need to consider the comments from Reviewer #1. A PDF file "PONE-D-20-04914 AE Comments" is required to be responded appropriately.

2. Please amend either the title on the online submission form (via Edit Submission) or the title in the manuscript so that they are identical.

4. Please remove your figures from within your manuscript file, leaving only the individual TIFF/EPS image files, uploaded separately.  These will be automatically included in the reviewers’ PDF.

Reviewers' comments:

Reviewer's Responses to Questions

**Comments to the Author**

1. Is the manuscript technically sound, and do the data support the conclusions?

Reviewer #1: Yes

Reviewer #2: Yes

2. Has the statistical analysis been performed appropriately and rigorously? 

Reviewer #1: Yes

Reviewer #2: I Don't Know

3. Have the authors made all data underlying the findings in their manuscript fully available?

Reviewer #1: Yes

Reviewer #2: Yes

4. Is the manuscript presented in an intelligible fashion and written in standard English?

Reviewer #1: Yes

Reviewer #2: Yes

5. Review Comments to the Author

Reviewer #1: I have read the paper with interest.

The manuscript topic is of interest for “PLOS ONE”.

The authors offer a work that potentially can improve our knowledge about this topic. I suggest that the authors consider a revision of their work along the following suggestions and questions.

Reviewer #2: The paper has investigated a new air-assited MEOR process. The paper proves a good addition in literature and add good addition in the field or EOR. Paper can be accepted after following revisions.

1. In the introduction part author has discussed about the background and concepts of MEOR process. The author should write in separate paragraphs, also add introduction about simulation part, and in the last paragraph discuss the summary of methodology in a bit more detail.

Add some of the recent articles about EOR, such as,

https://doi.org/10.1016/j.jclepro.2020.120777

https://doi.org/10.1016/j.molliq.2020.113095

2. Check for grammar and spelling mistakes.

3. In section 2.1, “The cultures were stored at a temperature of 4 °C for later use” why this temperature was choosen? Discuss and provide reference if necessary,

4. If methodology in section 2.3 was adopted from some other research provide references, otherwise provide reasoning.

5. Equation numbers should be in line, also provide references for all the equations which are used from some other research.

6. In table 3, the values should be in line

7. In Table 6, author has proposed different recovery rates because of different injection patterns, author has not discussed the effect of permeability, porosity and pressure difference on the recoevery rate. As these values are not constant, so author should take in account and discuss these parameters.

8. In Figure 9, captions and numbering should be placed properly.

9. Check format for equations 14 and 15

6. PLOS authors have the option to publish the peer review history of their article (what does this mean?). If published, this will include your full peer review and any attached files.

Reviewer #1: No

Reviewer #2: **Yes: **Associate Professor Dr. Hassan Soleimani

---

## [Author Response · Author response to Decision Letter 0]

3 Oct 2020

Reviewer #1: I have read the paper with interest.

The manuscript topic is of interest for “PLOS ONE”.

The authors offer a work that potentially can improve our knowledge about this topic. I suggest that the authors consider a revision of their work along the following suggestions and questions.

1- It is suggested to discuss more about the findings of this study in the abstract.

Response: More about the findings of this study was discussed in the abstract. The experiment shows that water flooding, microbial flooding and air - assisted microbial flooding core flow experiments were carried out. Carbon distribution curve of biodegraded oil with different oxygen concentration was determined by chromatographic analysis. The long-chain alkanes are degraded by microorganisms. This process is very conducive to lowering the oil viscosity.

2- It is recommended to mention about the applications of this study at the end of the abstract.

The findings of this study can help for better understanding of …

Response: The applications of this study were mentioned about at the end of the abstract.

The findings of this study can help for better understanding of microbial enhanced oil recovery and improving the efficiency of microbial oil displacement.

3- I strongly recommend the authors to add one paragraph discussing the difference between their work and the previously performed studies in literature. In other words, what is the novelty of this work? I offer the authors to revise the abstract and introduction in order to incorporate the novelty of their work. This change motivates the readers of “PLOS ONE” to study this work with interest.

Response: The abstract and introduction were revised in order to incorporate the novelty of our work.

The microbial enhanced oil recovery faced with a problem that the inefficient reproduction of microorganisms in oxygen-deprived environments of the reservoir. To overcome this problem, a new type of air-assisted MEOR process was investigated.

4- The following relevant references can be cited in introduction where “At the same time. non-renewable resources, such as oil, are at risk of depletion, consequently, applying enhanced oil recovery (EOR) techniques in developed fields is crucial to the maximization of scarce oil resources.” is addressed:

• (2020), Experimental study on the viscosity behavior of silica nanofluids with different ions of electrolytes. Industrial & Engineering Chemistry Research, 59(8): 3575-3583.

• (2020), Interfacial energy for solutions of nanoparticles, surfactants, and electrolytes. American Institute of Chemical Engineers Journal, 66(4): e16891.

2019), A comprehensive review on interaction of nanoparticles with low salinity water and surfactant for enhanced oil recovery in sandstone and carbonate reservoirs. Fuel, 241: 1045-1057.

• (2019), Mathematical modelling of surface tension of nanoparticles in electrolyte solutions. Chemical Engineering Science, 197: 345-356.

Response: The following relevant references were cited in introduction.

5- It is suggested to add a figure in section 1 (Introduction) which shows the general sketch of the problem.

Response: A figure was added in section 1 (Introduction) to show the general sketch of this study.

6- It is recommended to discuss about the “Recent Theoretical, Field, and Experimental Studies” in the introduction.

Response: The introduction was modified, the “Recent Theoretical, Field, and Experimental Studies” was discussed in the introduction.

7- It is recommended to include a paragraph at the end of introduction to present the steps of the work like:

First, the materials and methods are presented. Then,…

Response: A paragraph at the end of introduction was added to present the steps of the work.

The steps of the work were as flows: First, the materials and methods are presented. Then, the strain was identified and the microbial oxygen consumption rate is determined. Then, the influence of dissolved oxygen concentration on the carbon number distribution of biodegraded oil is measured. Then, the physical simulation of air-assisted MEOR in heterogeneous porous media is carried out. Finally, the mathematical model of air-assisted MEOR is established, the air-assisted microbial flooding performance is predicted.

8- What are the advantages and disadvantages of this study? I recommend the authors to highlight this topic.

Response: The advantages and disadvantages of this study is low cost, powerful adaptability, simple construction, harmless reservoir and no-pollution. Furthermore, the air-assisted MEOR technology could overcome the inefficient reproduction of traditional MEOR in oxygen-deprived environments of the reservoir.

The disadvantage of this study is that continuous air injection into the reservoir comes with a risk of oxygen breakthrough. In such case, the oxygen concentration may be reduced to a safe level through microorganism growth.

9- It is recommended to keep the main governing equations in the text of manuscript and move the rest of equations to an appendix.

Response: The main governing equations in the study are reedited.

10- The title for the last section should be replaced by “Summary and Conclusions”.

The title for the last section has been replaced by “Summary and Conclusions”.

11- It is recommended to show the main remarks of this study in terms of bullets in last section (Summary and Conclusions).

Response: The main remarks of this study were shown in terms of bullets in last section (Summary and Conclusions).

12- It is suggested to add a nomenclature.

Response: The nomenclature in the study was rechecked and modified. Compared to traditional MEOR, The “air-assisted Microbial Enhanced Oil Recovery” is a new nomenclature.

Reviewer #2: The paper has investigated a new air-assited MEOR process. The paper proves a good addition in literature and add good addition in the field or EOR. Paper can be accepted after following revisions.

1. In the introduction part author has discussed about the background and concepts of MEOR process. The author should write in separate paragraphs, also add introduction about simulation part, and in the last paragraph discuss the summary of methodology in a bit more detail.

Add some of the recent articles about EOR, such as,

https://doi.org/10.1016/j.jclepro.2020.120777

https://doi.org/10.1016/j.molliq.2020.113095

Response: Introduction about simulation part was added, and in the last paragraph discuss the summary of methodology in a bit more detail. The recent articles about EOR were referenced.

2. Check for grammar and spelling mistakes.

Response: The grammar and spelling mistakes were checked and revised.

3. In section 2.1, “The cultures were stored at a temperature of 4 °C for later use” why this temperature was choosen? Discuss and provide reference if necessary.

Response: The cultures were stored at a temperature of 4 °C for later use, because the strain is dormant at this temperature. Therefore, 4°C is the optimum temperature for preserving the species.

4. If methodology in section 2.3 was adopted from some other research provide references, otherwise provide reasoning.

 Response: The experiment was designed according to the definition of microbial oxygen consumption. Microbial oxygen consumption is the amount of dissolved oxygen in water consumed by microbial reproduction and metabolism. Oxygen consumption rate is the rate at which microorganisms use up oxygen for respiration. The experiment is referred to a paper published by Lei Guanglun in 2016.

5. Equation numbers should be in line, also provide references for all the equations which are used from some other research.

 Response: Equation number was readjusted to be in line.

6. In table 3, the values should be in line.

 Response: Table 3 has been reedited to be case - sensitive. 

7. In Table 6, author has proposed different recovery rates because of different injection patterns, author has not discussed the effect of permeability, porosity and pressure difference on the recoevery rate. As these values are not constant, so author should take in account and discuss these parameters.

Response: Core flooding experiments were conducted using artificial cores, thus the parameters of artificial cores could not be exactly consistent. The permeability, porosity and pressure of artificial cores were as close as possible.

8. In Figure 9, captions and numbering should be placed properly.

 Response: Figure 9 contains four diagrams:（a）is the core flooding experiment of water driving，(b) is the core flooding experiment of microbial flooding, (c) is the core flooding experiment of air- microorganisms-air- microorganisms flooding,(d)is the core flooding experiment of microorganisms-air- microorganisms-air flooding. 

 9. Check format for equations 14 and 15.

 Response: Equations 14 and 15 have been reedited.

---

## [Editor Report · Decision Letter 1]

8 Oct 2020

PONE-D-20-04914R1

Separating and Evaluating Oil Degradation Strains for the Air-assisted Microbial Enhanced Oil Recovery Process

PLOS ONE

Dear Dr. Mingming Cheng,

Thank you for submitting your manuscript to PLOS ONE. After careful consideration, we feel that it has merit but does not fully meet PLOS ONE’s publication criteria as it currently stands. Therefore, we invite you to submit a revised version of the manuscript that addresses the points raised during the review process.

ACADEMIC EDITOR:

As commented to authors.

We look forward to receiving your revised manuscript.

Kind regards,

Omeid Rahmani

Academic Editor

PLOS ONE

Additional Editor Comments (if provided):

The authors have improved the manuscript "Separating and Evaluating Oil Degradation Strains for the Air-assisted Microbial Enhanced Oil Recovery Process" in somehow that is possible. However, there are some main points to be addressed before it can be proceeded further.

Lines 30-31: Incomplete statement. Check it: "The current rapid pace of economic and social development worldwide necessitates, ...."

Table 2: Determine how the parameters have analyzed? Which device or analysis is used to measure these parameters and elements?

"Transparent organic glass" Provide the due properties.

Line 117: sterilization for 30 min in high-pressure steam. At what level of pressure? Clarify.

Line 118: gas flow control devices. How many g-f devices are applied?

What is/are difference(s) between Lines 116-120 and 134-137? After sterilization .....for 5 days.

Line 154: discuss more about "Buckley-Leverett theory".

Fi. 5: Did you adapt Fig. 4 from Cheng et al. or others? If yes, provide its Rightslink with a proper citation.

Lines 270-271: "This strain has a strong ability to break down organics into molecules such as glycolipids and lipopeptides." Any reference?

Fig.7: An elucidation is required here to state about the used digits in Fig. 7. Please provide a sufficient explanation about digits used in Fig. 7, maybe as Supplementary Data.

Fig.8 (Right): It is not clear that the bars (y= -0.014x+0.702 ...) are related to which curve? Change color accordingly.

Table 5 should be discussed more in the text. What are the results and their application in the process of MEOR?

Line 323: "As the table shows" which table?

Fig. 10 a: Why the trend of water cut is dis-order? Clarify within the text briefly.

Remove below references from the list and also the text. They are irrelevant as recommended to the authors in the first stage:

Saheed O and Morteza D. Interfacial energy for solutions of nanoparticles, surfactants, and electrolytes[J]. American Institute of Chemical Engineers Journal, 2020, 66(4): e16891.

Saheed O and Morteza D. Experimental Study on the Viscosity Behavior of Silica Nanofluids with Different Ions of Electrolytes [J]. 2020, 59, 8, 3575–3583

---

## [Author Response · Author response to Decision Letter 1]

29 Nov 2020

The authors have improved the manuscript "Separating and Evaluating Oil Degradation Strains for the Air-assisted Microbial Enhanced Oil Recovery Process" in somehow that is possible. However, there are some main points to be addressed before it can be proceeded further.

Lines 30-31: Incomplete statement. Check it: "The current rapid pace of economic and social development worldwide necessitates, ...."

Lines 30-31 was modified to “The current rapid pace of economic and social development worldwide necessitates the use of ever-increasing amounts of resources (Cheng et al., 2014). ”

Table 2: Determine how the parameters have analyzed? Which device or analysis is used to measure these parameters and elements?

The parameters have been analyzed by chemical titration. 

Line 117: sterilization for 30 min in high-pressure steam. At what level of pressure? Clarify.

Line 117 was modified to “After sterilization for 30 min in high-pressure steam (0.12MPa), ”

Line 118: gas flow control devices. How many g-f devices are applied?

Line 117 was modified to “air was introduced into the first three flasks via a gas flow control devices to set the dissolved oxygen concentrations to the desired values of 3.0, 4.5, and 5.5 mg/L.”

What is/are difference(s) between Lines 116-120 and 134-137? After sterilization .....for 5 days.

 “while stirring continuously, for 5 days” is bacterial culture process. Lines 116-120 were bacterial culture for determination of microbial oxygen consumption rate, Lines 134-137 were bacterial culture for Influence of dissolved oxygen concentration on the carbon number distribution.

Line 154: discuss more about "Buckley-Leverett theory".

Line 154 was modified to “Under the precondition of incompressible flow, the capillary pressure and gravity are not considered，the mathematical model used in this study is based on the Buckley-Leverett theory and Darcy’s law.”

Fi. 5: Did you adapt Fig. 4 from Cheng et al. or others? If yes, provide its Rightslink with a proper citation.

A sentence was added “Concrete method for oxygen-consumption model was shown in Fig.5( Cheng (Cheng et al., 2014).” before Fig.5. 

Lines 270-271: "This strain has a strong ability to break down organics into molecules such as glycolipids and lipopeptides." Any reference?

Reference was add in Lines 270-271.

Fig.7: An elucidation is required here to state about the used digits in Fig. 7. Please provide a sufficient explanation about digits used in Fig. 7, maybe as Supplementary Data.

The digits represent the similarity of the strains to the database cultures. 

Fig.8 (Right): It is not clear that the bars (y= -0.014x+0.702 ...) are related to which curve? Change color accordingly.

An indicator line indicates that the relationship of the bars (y= -0.014x+0.702 ...)and the curve.

Table 5 should be discussed more in the text. What are the results and their application in the process of MEOR?

Some sentences were added here. Table 5 shows that the oxygen consumption rate of microorganism is different with different oxygen content. This is because the metabolic rate of microorganisms was different under different oxygen content. The oxygen concentration distribution was calculated using the data of microbial metabolic rate.

Line 323: "As the table shows" which table?

This line was changed to “As Table 6 shows,”

Fig. 10 a: Why the trend of water cut is dis-order? Clarify within the text briefly.

Some sentences were added here. Although the water cut fluctuates in some moment, it shows a larger upward trend. The water cut curves of air-assisted microbial flooding have obvious fall during secondary water drive stage. It shows that the air-assisted microbial flooding is effective in water cut control during enhanced oil recovery process.

Remove below references from the list and also the text. They are irrelevant as recommended to the authors in the first stage:

Saheed O and Morteza D. Interfacial energy for solutions of nanoparticles, surfactants, and electrolytes[J]. American Institute of Chemical Engineers Journal, 2020, 66(4): e16891.

Saheed O and Morteza D. Experimental Study on the Viscosity Behavior of Silica Nanofluids with Different Ions of Electrolytes [J]. 2020, 59, 8, 3575–3583

The below references were remove from the list and also the text.

---

## [Editor Report · Decision Letter 2]

2 Dec 2020

Isolating, Identifying and Evaluating of Oil Degradation Strains for the Air-assisted microbial Enhanced Oil Recovery Process

PONE-D-20-04914R2

Dear Dr. Cheng,

We’re pleased to inform you that your manuscript has been judged scientifically suitable for publication and will be formally accepted for publication once it meets all outstanding technical requirements.

Kind regards,

Omeid Rahmani

Academic Editor

PLOS ONE
---

## [Editor Report · Acceptance letter]

2 Jan 2021

PONE-D-20-04914R2 

Isolating, Identifying and Evaluating of Oil Degradation Strains for the Air-assisted microbial Enhanced Oil Recovery Process 

Dear Dr. Cheng:

I'm pleased to inform you that your manuscript has been deemed suitable for publication in PLOS ONE. Congratulations! Your manuscript is now with our production department. 

Kind regards, 

on behalf of

Dr. Omeid Rahmani 

Academic Editor

PLOS ONE